

# Parental encouragement is positively associated with outdoor active play outside of school hours among 7–12 year olds

Thomas Ferrao[1] and Ian Janssen[1,2]

[1] School of Kinesiology and Health Studies, Queen's University, Kingston, ON, Canada
[2] Department of Public Health Sciences, Queen's University, Kingston, ON, Canada

## ABSTRACT

**Background.** Physical activity is important for children's physical, mental, and social well-being. Outdoor active play is an important yet unstudied domain of children's physical activity. The objective of this study was to determine if parental encouragement is associated with the frequency that children engage in outdoor active play outside of school hours.

**Methods.** Participants consisted of 514 children aged 7–12 years and one of their parents. Parents completed a survey that included four questions that assessed how frequently they used verbal cues to encourage their child to play outdoors. Points were assigned to each response and averaged across the 4 questions, and based on this average participants were assigned to quintiles. The survey included seven questions that asked parents to assess how frequently their child played outdoors outside of school hours. Points were assigned to each response and summed to create an active outdoor play frequency score. General linear models assessed associations between parental encouragement and outdoor play while controlling for individual, family, and neighborhood covariates.

**Results.** The mean outdoor active play frequency score increased significantly across quintiles of the parental encouragement score as follows: 6.0 (standard error = 0.7) in quintile 1, 9.8 (0.6) in quintile 2, 11.4 (0.6) in quintile 3, 16.2 (0.9) in quintile 4, and 23.3 (1.3) in quintile 5. After adjusting for covariates, the mean outdoor active play frequency score was almost three times higher in the highest parental encouragement quintile than in the lowest quintile (20.4 vs. 7.8).

**Conclusions.** Parents use of verbal cues to encourage their children to play outdoors was independently associated with outdoor active play among 7–12 year olds.

Corresponding author
Ian Janssen, ian.janssen@queensu.ca

## INTRODUCTION

Regular physical activity is important for children's health (*Janssen & Leblanc, 2010*). It is recommended that 5–17 year old children accumulate at least 60 min of moderate-to-vigorous physical activity (MVPA) daily (*Tremblay et al., 2011*). Worryingly, objective physical activity measures suggest that the majority of children in the United States and

other developed countries do not meet this recommendation (*Colley et al., 2011*; *Hallal et al., 2012*; *Troiano et al., 2008*). The primary domains in which children accumulate their MVPA are by participating in organized sport (e.g., soccer game, swimming lessons, dance class), physical education class and other structured school activities, active transport (e.g., walking and cycling), and active play (e.g., tag, road hockey, hide and seek) (*Veitch, Salmon & Ball, 2010*). Of these four domains, it has been argued that active play is the domain where children are performing the poorest (*Active Healthy Kids Canada, 2012*). This is concerning because in addition to influencing physical health, active play provides cognitive, social, and emotional benefits through the development of creativity, problem solving, and conflict resolution (*Brockman, Jago & Fox, 2011*).

A limited amount of research has examined the determinants of the active play domain of physical activity. One potential determinant is parental support, and an important aspect of parental support is encouragement (*Gustafson & Rhodes, 2006*). In the context of active play, encouragement could consist of parents giving verbal cues for their child to engage in play such as telling them to go to the park or to ride their bicycle with friends in the neighborhood. Although several studies have shown that parental encouragement is associated with total MVPA among children (*Anderssen & Wold, 1992*; *Biddle & Goudas, 1996*; *McGuire et al., 2002*; *Pugliese & Tinsley, 2007*; *Welk, Wood & Morss, 2003*), we are aware of only a single study which has linked parental encouragement and the active play domain of physical activity. That study reported that the relative odds of playing active games most of the time during the school recess period was 0.70 (95% CI [0.46–1.04]) in 12–14 year old children who received low parental encouragement by comparison to children who received high parental encouragement (*Hohepa et al., 2007*). The influence of parental encouragement on active play outside of school settings has not been established. This is an important gap in the literature. Parents have limited control of their children during the school day and should have a much greater influence on their child's behaviors outside of school hours.

The purpose of this study was to determine whether parental encouragement for outdoor active play is associated with outdoor active play performed outside school hours within 7–12 year olds. It was hypothesized that parental encouragement would be positively associated with outdoor active play. The findings from this study could direct future interventions and could provide parents with insights on approaches they could use to improve the physical activity levels and ultimately the health of their children.

## MATERIALS AND METHODS

### Study design and participants

This was a cross-sectional study of children born between 2003 and 2007 and aged 7–12 years at the time data were collected. The study received ethic clearance from the Queen's University General Research Ethics Board (file # 6014210). Data was obtained through questionnaire via parent/guardian proxy report. Parents were recruited from the CINT panel, a heterogeneous group of over 15 million adults from over 60 countries who participate in web-based surveys. Inclusion criteria stipulated that the panel member speak

English, reside in the United States, and have at least one child of the appropriate age. If the panel member had more than one 7–12 year old child, they were asked to complete the survey based on the oldest of those children. A single panel member per internet protocol (IP) address was allowed to participate. Panel members were required to read the letter of information and provide consent before completing the survey, which was administered using FluidSurveys™ online survey software. All surveys were completed on November 24, 2014. Altogether, 514 parents/guardians completed the survey. The majority of parent respondents were female (75%) and a biological parent of the child (92%).

## Parental encouragement

Parental encouragement was measured using items that were developed through focus group testing with parents (*Davison et al., 2011*). Parents were asked the extent to which they encouraged their child's outdoor active play through the following statements: (1) "I encourage my child to use resources in our neighborhood to be active (such as the park and the school)," (2) "I encourage my child to walk or ride his/her bike in our neighborhood if it is safe and appropriate for his/her age," (3) "I encourage my child to play outdoors (without adult supervision) when the weather is nice," and (4) "I encourage my child to play outdoors in our yard and/or driveway." Response options (and corresponding point allotments) were as follows: "never/rarely" (0), "less than once a week" (0.5), "1–2 times per week" (1.5), "3–4 times per week" (3.5), "5–6 times per week" (5.5), and "daily" (7). Points were averaged across the 4 questions, and based on this average participants were assigned to quintiles. The range of points for each quintile was as follows: Q1 (0–1.4), Q2 (1.5–2.5), Q3 (2.6–3.6), Q4 (3.8–5), and Q5 (5.3–7).

## Outdoor active play

The frequency children engaged in outdoor active play was measured using a previously developed item (*Veitch, Salmon & Ball, 2009*). Parents were asked the following question: "Thinking about the past month, in a usual week how often did your child play outdoors in the following locations?" The seven locations included "the yard at your home," "the yard at someone else's home (friend, neighbor or relative)," "the street or cul-de-sac your home is on," "other streets or cul-de-sacs," "parks and playgrounds outside of school hours," "school grounds outside of school hours," and "other places where your child can be active (e.g., field, parking lot, forested area)." The following six response options (and corresponding points allocation) were provided for each location: "never/rarely" (0), "less than once a week" (0.5), "1–2 times per week" (1.5), "3–4 times per week" (3.5), "5–6 times per week" (5.5), and "daily" (7). The points allotted for these items were summed to create an outdoor active play frequency score that had a potential range of 0 to 49. Intra-class correlation coefficients from the two week test-retest reliability of these items range from 0.58 to 0.82 (*Veitch, Salmon & Ball, 2009*).

## Total physical activity outside of school

Parents were asked the weekly frequency during the past month that their child walked or bicycled to 8 different locations (*Timperio et al., 2004*). Parents were also asked how

frequently their child participated in organized sport. The response options and point allotments for active transportation and organized sport were the same as those described above for outdoor active play. The frequency of all major physical activity domains performed outside of school hours was determined by summing items addressing outdoor active play, active transportation, and organized sport.

## Covariates

Potential covariates included child and family demographics such as gender, age, race (Non-Hispanic White, Non-Hispanic Black, Hispanic, and Other including mixed race), parental structure (dual parent or single parent home), number of siblings (0, 1, 2, and 3 or more), annual household income ($\leq$$25,000, $25,001–$50,000, $50,001–$75,000, $75,001–$100,000, and $\geq$$100,001), and highest education of the parent completing the survey (high school or less, 2 year college, 4 year college/university, and graduate university or higher).

Other parental influences on active play were also considered. These consisted of parental facilitation, involvement, modelling for physical activity, and the child's independent mobility. Parental facilitation (1 item), involvement (2 items), and modelling (3 items) were measured using items from the Activity Support Scale for Multiple Groups (ACTS-MG) with response options ranging from "strongly disagree" (0 points) to "strongly agree" (4 points) (*Davison et al., 2011*). When assessed using more than one item, the points were averaged and these scores were inserted into the regression models as continuous variables. Independent mobility was measured using items that asked how far from home the child was allowed to roam unsupervised, with responses ranging from "my child is not allowed out alone" to "my child is allowed out more than a 15 min walk from home" (*Veitch et al., 2014*).

Finally, neighborhood and community factors were considered. We inquired about the population size of the municipality where participants lived ($\leq$9,999, 10,000–99,999, 100,000–499,999, $\geq$500,000 people) and the form or neighborhood they lived in (rural, semi-rural, suburban, urban). We also asked 18 questions around parents' perceptions of safety in their neighborhood (*Carver, Timperio & Crawford, 2008a*). Principal component analysis with an oblique rotation was performed to reduce these 18 items. Four factors emerged, with 4–5 questions being included in each factor. We labelled these factors as follows: unsafe roads ($\alpha = 0.81$, $\lambda = 3.01$), traffic calming ($\alpha = 0.65$, $\lambda = 2.03$), safe for children ($\alpha = 0.80$, $\lambda = 2.95$), and crime risk ($\alpha = 0.76$, $\lambda = 2.51$). Anderson–Rubin scores were computed for each factor and these scores were included in the regression models as continuous variables.

## Statistical analysis

Analyses were conducted in IBM SPSS version 22. Conventional descriptive statistics were used to describe the sample. General linear models were used to analyze the relationship between parental encouragement and the covariates with outdoor active play. Initially we conducted a series of bivariate analyses. This was followed by a multivariate model that simultaneously included the primary exposure (parental encouragement) and all of the

covariates. Next, we used a backwards elimination approach to remove covariates from the multivariate model that had a $p$ value of $\geq 0.1$. These regression analyses were repeated using the frequency of total physical activity outside of school as the outcome.

In order to detect a medium effect size with a power level of 0.9 and a significance level of $\alpha = .005$ to account for multiple group comparisons, a minimum sample size of 99 participants per group was required. Since we planned on making comparisons across 5 equally size groups (i.e., quintiles) for the primary exposure variable, a total sample of 500 was targeted.

## RESULTS

Descriptive statistics for the 514 children are in Table 1. Approximately half were male (49%) and lived in suburbs or subdivisions (47%). The majority were non-Hispanic White (69%) and lived in a dual parent household (85%). Table 2 shows the distribution of responses to the parental encouragement questions. Approximately one third of parents reported that they encouraged their child to play outdoors when the weather is nice and to play in their yard/driveway on a daily basis. Approximately 13% encouraged their child to be active by using resources in the neighborhood and to walk/bike in their neighborhood on a daily basis. Table 3 shows the distribution of responses to the items addressing the frequency of outdoor active play. Approximately 22% of parents reported that their child played in the yard on a daily basis. For the other 6 outdoor play locations, <10% of parents reported that their child played there on a daily basis.

The mean outdoor active play frequency score within the entire sample was 13.3 (SE = 0.5). There was a significant increase in this score across parental encouragement quintiles as follows: 6.0 (0.7) in quintile 1, 9.8 (0.6) in quintile 2, 11.4 (0.6) in quintile 3, 16.2 (0.9) in quintile 4, and 23.3 (1.3) in quintile 5. As shown in Fig. 1, after adjusting for covariates there was an almost threefold difference in the mean outdoor active play frequency score when comparing the lowest and highest parental encouragement quintiles (7.8 vs. 20.4). Associations between parental encouragement and outdoor active play are further shown in Table 4. The final multivariate regression model indicated that there was a 12.7 (1.2) point difference in the outdoor play frequency score between the lowest and highest parental encouragement quintiles. Covariates retained in the final multivariate model were the number of siblings, parental involvement, and the unsafe roads, traffic calming, safe for children, and crime risk factors.

The mean frequency score for total physical activity outside of school (outdoor active play + active transportation + organized sport) within all 514 children was 24.3 (SE = 1.0). This score increased significantly across parental encouragement quintiles as follows: 10.5 (1.1) in quintile 1, 18.8 (1.3) in quintile 2, 19.5 (1.5) in quintile 3, 28.7 (2.1) in quintile 4, and 43.7 (2.9) in quintile 5. The final multivariate regression model indicated that there was a 21.1 (2.4) point difference in the total physical activity frequency score between the lowest and highest parental encouragement quintiles (Table 5). Age, parental structure, number of siblings, education of the parent completing the survey, independent

**Table 1 Descriptive information of children (N = 514).**

| Variable | N | % |
|---|---|---|
| *Gender* | | |
| Male | 251 | 48.8 |
| Female | 263 | 51.2 |
| *Age (years)* | | |
| 7–8 | 173 | 33.7 |
| 9–10 | 222 | 43.2 |
| 11–12 | 119 | 23.2 |
| *Race* | | |
| Non-hispanic white | 355 | 69.1 |
| Non-hispanic black | 32 | 6.2 |
| Hispanic | 79 | 15.4 |
| Other | 48 | 9.3 |
| *Parental structure* | | |
| Dual parent | 417 | 81.1 |
| Single parent | 97 | 18.9 |
| *Number of siblings in household* | | |
| 0 | 124 | 24.1 |
| 1 | 206 | 40.1 |
| 2 | 119 | 23.2 |
| 3 or more | 65 | 12.6 |
| *Household income ($ per year)* | | |
| $\leq$25,000 | 77 | 15.0 |
| 25,001–50,000 | 119 | 23.2 |
| 50,001–75,000 | 106 | 20.6 |
| 75,001–100,000 | 114 | 22.2 |
| $\geq$100,001 | 98 | 19.1 |
| *Parental education* | | |
| High school or less | 118 | 23.0 |
| 2-year college | 132 | 25.7 |
| 4-year college/university | 193 | 37.5 |
| Graduate university | 71 | 13.8 |
| *Population size of municipality* | | |
| $\leq$9,999 | 123 | 23.9 |
| 10,000–99,999 | 136 | 26.5 |
| 100,000–499,999 | 127 | 24.7 |
| $\geq$500,000 | 128 | 24.9 |
| *Urban form* | | |
| Rural | 84 | 16.3 |
| Semi-rural | 84 | 16.3 |
| Suburb or subdivision | 243 | 47.3 |
| Urban or inner-city | 103 | 20.0 |

**Table 2 Frequency of parental encouragement for outdoor active play (N = 514).**

| Parents encourage child to... | Weekly frequency of encouragement | | | | | |
|---|---|---|---|---|---|---|
| | Never/rarely (%) | <Once (%) | 1–2 times (%) | 3–4 times (%) | 5–6 times (%) | Daily (%) |
| Use resources in neighbourhood to be active | 23.9 | 15.4 | 26.1 | 14.8 | 7.4 | 12.5 |
| Walk or bike in neighbourhood if it is safe | 24.9 | 10.9 | 25.3 | 18.9 | 7.2 | 12.8 |
| Play outdoors when the weather is nice | 4.1 | 5.1 | 19.5 | 22.2 | 13.8 | 35.4 |
| Play outdoors in their yard and/or driveway | 5.3 | 5.8 | 20.6 | 20.2 | 15.0 | 33.1 |

**Table 3 Children's frequency of outdoor active play at different locations (N = 514).**

| Outdoor active play location | Weekly Frequency of Participation | | | | | |
|---|---|---|---|---|---|---|
| | Never/rarely (%) | <Once (%) | 1–2 Times (%) | 3–4 Times (%) | 5–6 Times (%) | Daily (%) |
| Yard at home | 6.4 | 9.3 | 24.3 | 26.1 | 12.3 | 21.6 |
| Yard at someone else's home | 21.2 | 21.0 | 30.4 | 14.2 | 7.6 | 5.6 |
| Street or cul-de-sac home is on | 43.0 | 15.6 | 16.1 | 11.7 | 6.4 | 7.2 |
| Other streets or cul-de-sacs | 63.8 | 14.6 | 9.7 | 5.6 | 2.7 | 3.5 |
| Parks and playgrounds outside of school hours | 15.4 | 30.5 | 28.0 | 13.4 | 8.0 | 4.7 |
| School grounds outside of school hours | 39.1 | 15.8 | 12.5 | 10.7 | 13.6 | 8.4 |
| Other places (e.g., field, parking lot, forested area) | 21.4 | 25.1 | 27.8 | 13.2 | 7.6 | 4.9 |

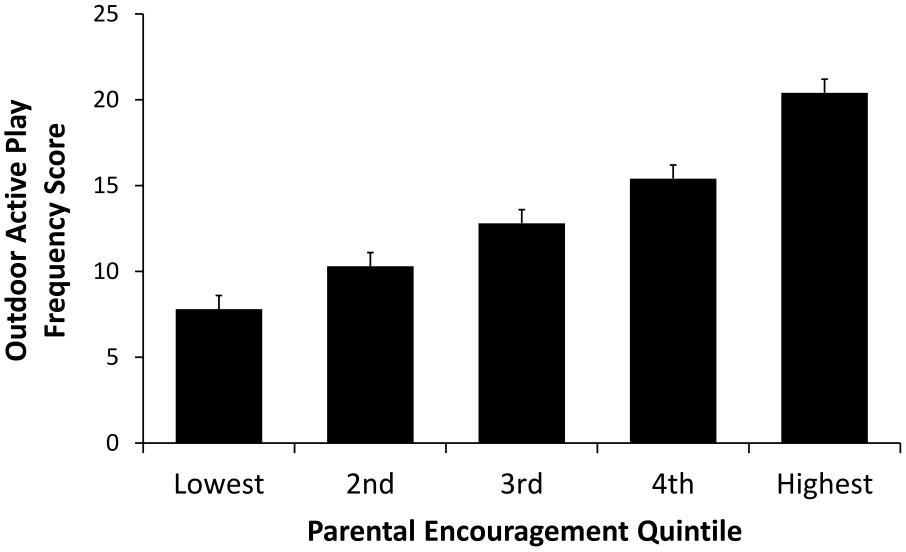

**Figure 1 Mean outdoor active play frequency scores within parental encouragement quintiles.** Means were adjusted for number of siblings, parental involvement, and neighbourhood safety factors (unsafe roads, traffic calming, safe for children, crime risk). Error bars represent standard errors.

**Table 4  Association between parental encouragement for outdoor active play and covariates with the frequency of outdoor active play.**

| | Bivariate models, $\beta$ (S.E.) | Multivariate model with all covariates, $\beta$ (S.E.) | Multivariate model with significant covariates, $\beta$ (S.E.) |
|---|---|---|---|
| *Parental encouragement* | | | |
| Quintile 1 (lowest) | 0 | 0 | 0 |
| Quintile 2 | 3.8 (1.2) | 2.3 (1.1) | 2.5 (1.1) |
| Quintile 3 | 5.4 (1.2) | 4.4 (1.2) | 5.0 (1.1) |
| Quintile 4 | 10.2 (1.3) | 7.3 (1.2) | 7.6 (1.2) |
| Quintile 5 (highest) | 17.3 (1.2) | 12.0 (1.2) | 12.7 (1.2) |
| *Gender* | | | |
| Male | 0 | 0 | |
| Female | −2.6 (0.9) | −0.8 (0.7) | |
| *Age (years)* | | | |
| 7–8 | 0 | 0 | |
| 9–10 | −2.2 (1.1) | −1.1 (0.8) | |
| 11–12 | −2.9 (1.3) | −2.2 (1.0) | |
| *Race* | | | |
| Non-hispanic white | 0 | 0 | |
| Non-hispanic black | −0.0 (2.0) | −0.8 (1.5) | |
| Hispanic | 2.1 (1.3) | 1.3 (1.0) | |
| Other | −2.0 (1.6) | 0.1 (1.2) | |
| *Parental structure* | | | |
| Dual parent | 0 | 0 | |
| Single parent | 0.8 (1.2) | 1.1 (1.0) | |
| *Number of siblings in household* | | | |
| 0 | 0 | 0 | 0 |
| 1 | −2.2 (1.2) | −0.5 (0.9) | −0.6 (1.2) |
| 2 | −3.2 (1.4) | −2.9 (1.0) | −3.4 (1.0) |
| 3 or more | −1.5 (1.6) | −0.2 (1.2) | −1.1 (1.2) |
| *Household income ($ per year)* | | | |
| ≤25,000 | 0 | 0 | |
| 25,001–50,000 | −1.2 (1.5) | −0.0 (1.2) | |
| 50,001–75,000 | −0.6 (1.6) | −1.0 (1.3) | |
| 75,001–100,000 | 0.5 (1.6) | −0.4 (1.4) | |
| ≥100,001 | 1.6 (1.6) | 0.3 (1.4) | |
| *Parental education* | | | |
| High school or less | 0 | 0 | |
| 2 year college | −0.7 (1.3) | −0.0 (1.0) | |
| 4 year college/university | 1.8 (1.2) | 1.4 (1.0) | |
| Graduate university | 3.2 (1.6) | 1.2 (1.3) | |
| *Independent mobility* | | | |
| Not allowed out | 0 | 0 | |
| Within own yard | −0.6 (1.5) | −0.6 (1.2) | |
| Within own street | 1.2 (1.6) | −0.5 (1.3) | |

Table 4 (*continued*)

| | Bivariate models, β (S.E.) | Multivariate model with all covariates, β (S.E.) | Multivariate model with significant covariates, β (S.E.) |
|---|---|---|---|
| Within 2–3 streets | 3.8 (1.7) | 1.4 (1.4) | |
| >3 streets | 5.6 (2.0) | 0.9 (1.7) | |
| *Parental facilitation score* | 3.1 (0.5) | 0.3 (0.5) | |
| *Parental involvement score* | 3.6 (0.6) | 0.7 (0.7) | 1.1 (0.5) |
| *Parental modelling score* | 3.1 (0.5) | 0.2 (0.5) | |
| *Population size of municipality* | | | |
| ≤9,999 | 0 | 0 | |
| 10,000–99,999 | −0.2 (1.3) | −0.6 (1.1) | |
| 100,000–499,999 | 0.7 (1.3) | −0.2 (1.1) | |
| ≥500,000 | 3.7 (1.3) | 0.8 (1.2) | |
| *Urban form* | | | |
| Rural | 0 | 0 | |
| Semi-rural | 1.4 (1.6) | −0.1 (1.3) | |
| Suburbs | 0.0 (1.3) | −1.4 (1.2) | |
| Urban | 5.1 (1.5) | −1.0 (1.4) | |
| *Unsafe roads factor* | 1.1 (0.5) | 0.8 (0.4) | 0.9 (0.4) |
| *Traffic calming factor* | 3.8 (0.4) | 0.8 (0.5) | 1.0 (0.4) |
| *Safe for children factor* | 4.8 (0.4) | 2.9 (0.5) | 3.1 (0.4) |
| *Crime risk factor* | 2.6 (0.5) | 1.8 (0.5) | 1.7 (0.4) |

mobility, parental facilitation, and the unsafe roads, traffic calming, safe for children, and crime risk factors were the covariates retained in the final multivariate model.

## DISCUSSION

The key finding of this study is that parental encouragement was a strong, positive correlate of outdoor active play. In fact, of all the variables examined in the multivariate model, parental encouragement was the most strongly associated with the frequency of outdoor play. After adjusting for covariates, children who were encouraged the most played outdoors three times more frequently than children who were encouraged the least. Parental encouragement for outdoor active play was similarly associated with total physical activity outside of school hours, which implies that encouragement for and participation in outdoor active play did not negatively influence other domains of physical activity.

Consistent with our findings for active play, previous studies found that parental encouragement is associated with child's total physical activity (*Anderssen & Wold, 1992*; *Biddle & Goudas, 1996*; *Pugliese & Tinsley, 2007*; *Welk, Wood & Morss, 2003*). Furthermore, *Hohepa et al. (2007)* found that perceived encouragement by parents was associated with active games played during school recess among 12–14 year olds. This is an indicator of active play because recess activities are highly unstructured and self-directed by children. However, school recess is supervised by teachers and/or other caregivers and not under parental control, and therefore there may not be a direct casual association between parental encouragement and play during recess. Our study focused on the role of parent encouragement on active play outside of the school setting and this would

**Table 5** Association between parental encouragement for outdoor active play and covariates with the frequency of total physical activity (outdoor active play + active transportation + organized sport) outside of school.

| | Bivariate models, $\beta$ (S.E.) | Multivariate model with all covariates, $\beta$ (S.E.) | Multivariate model with significant covariates, $\beta$ (S.E.) |
|---|---|---|---|
| *Parental encouragement* | | | |
| Quintile 1 (lowest) | 0 | 0 | 0 |
| Quintile 2 | 8.3 (2.7) | 4.8 (2.3) | 4.8 (2.2) |
| Quintile 3 | 9.0 (2.6) | 6.9 (2.3) | 7.7 (2.2) |
| Quintile 4 | 18.1 (2.7) | 12.0 (2.4) | 12.3 (2.3) |
| Quintile 5 (highest) | 33.2 (2.6) | 20.8 (2.5) | 21.1 (2.4) |
| *Gender* | | | |
| Male | 0 | 0 | |
| Female | −4.6 (1.9) | −0.6 (1.4) | |
| *Age (years)* | | | |
| 7–8 | 0 | 0 | 0 |
| 9–10 | −5.3 (2.2) | −2.5 (1.7) | −2.6 (1.6) |
| 11–12 | −5.5 (2.6) | −3.9 (2.0) | −4.2 (1.9) |
| *Race* | | | |
| Non-hispanic white | 0 | 0 | |
| Non-hispanic black | 4.0 (4.0) | 2.0 (3.0) | |
| Hispanic | 5.5 (2.7) | 2.6 (2.0) | |
| Other | −3.6 (3.4) | −0.5 (2.5) | |
| *Parental structure* | | | |
| Dual parent | 0 | 0 | 0 |
| Single parent | 2.8 (2.5) | 2.8 (2.0) | 3.3 (1.9) |
| *Number of siblings in household* | | | |
| 0 | 0 | 0 | 0 |
| 1 | −6.3 (2.5) | −2.5 (1.8) | −2.7 (1.8) |
| 2 | −7.8 (2.8) | −7.1 (2.1) | −7.1 (2.1) |
| 3 or more | −4.9 (3.4) | −1.7 (2.5) | −1.4 (2.5) |
| *Household income ($ per year)* | | | |
| ≤25,000 | 0 | 0 | |
| 25,001–50,000 | −1.2 (3.2) | 1.9 (2.4) | |
| 50,001–75,000 | 0.6 (3.3) | −0.6 (2.6) | |
| 75,001–100,000 | 1.9 (3.3) | −0.9 (2.7) | |
| ≥100,001 | 5.6 (3.4) | 1.7 (2.9) | |
| *Parental education* | | | |
| High school or less | 0 | 0 | 0 |
| 2 year college | −1.9 (2.8) | −1.2 (2.0) | −1.5 (2.0) |
| 4 year college/university | 5.6 (2.5) | 4.4 (2.0) | 4.3 (1.9) |
| Graduate university | 9.6 (3.3) | 4.5 (2.7) | 4.7 (2.4) |
| *Independent mobility* | | | |
| Not allowed out | 0 | 0 | 0 |
| Within own yard | −3.0 (3.2) | −1.7 (2.4) | −1.9 (2.3) |

Table 5 (*continued*)

| | Bivariate models, $\beta$ (S.E.) | Multivariate model with all covariates, $\beta$ (S.E.) | Multivariate model with significant covariates, $\beta$ (S.E.) |
|---|---|---|---|
| Within own street | 1.1 (3.2) | −2.2 (2.5) | −2.3 (2.5) |
| Within 2–3 streets | 6.0 (3.5) | 1.5 (2.8) | 1.1 (2.8) |
| >3 streets | 13.1 (4.3) | 3.9 (3.4) | 4.8 (3.3) |
| *Parental facilitation score* | 6.5 (1.0) | 0.9 (1.0) | 1.5 (0.8) |
| *Parental involvement score* | 7.1 (1.2) | 0.5 (1.3) | |
| *Parental modelling score* | 6.8 (1.0) | 0.6 (1.0) | |
| *Population size of municipality* | | | |
| ≤9,999 | 0 | 0 | |
| 10,000–99,999 | −1.0 (2.7) | −1.7 (2.2) | |
| 100,000–499,999 | 3.5 (2.7) | 0.7 (2.3) | |
| ≥500,000 | 10.5 (2.7) | 2.0 (2.5) | |
| *Urban form* | | | |
| Rural | 0 | 0 | |
| Semi-rural | 2.0 (3.3) | −1.5 (2.5) | |
| Suburbs | 1.3 (2.7) | −3.6 (2.3) | |
| Urban | 14.0 (3.2) | −1.7 (2.8) | |
| *Unsafe roads factor* | 2.7 (1.0) | 1.8 (0.8) | 1.8 (0.8) |
| *Traffic calming factor* | 9.8 (0.9) | 2.9 (0.9) | 2.9 (0.9) |
| *Safe for children factor* | 10.9 (0.8) | 7.0 (1.0) | 7.1 (0.9) |
| *Crime risk factor* | 6.9 (0.9) | 4.3 (0.9) | 4.5 (0.9) |

have included outdoor active play that was supervised by parents and unsupervised play that was allowed by parents. Our findings suggest that the positive association between parental encouragement and outdoor active play is not a function of children substituting active play for other domains of physical activity such as organized sport and active transportation. Thus, parental encouragement for active play should benefit the child's total physical activity level.

Parental encouragement for outdoor active play in the form of simple verbal cues could influence children's physical activity in a number of ways. It could have a direct influence by inspiring children to spend their free time participating in active play rather than in sedentary pursuits such as screen time. It could indirectly influence active play by impacting their child's self-efficacy (*Gustafson & Rhodes, 2006*; *Sallis et al., 1992*), perceived competence (*Biddle & Goudas, 1996*; *Brustad, 1993*; *Welk, Wood & Morss, 2003*), attitudes (*Sallis, Prochaska & Taylor, 2000*; *Salmon et al., 2005*) and beliefs (*Heitzler et al., 2006*), all of which are known determinants of physical activity.

Although intervention studies are needed, the results from this study suggest that getting parents to provide more encouragement for active play would be an effective strategy for increasing children's active play levels. Interventions may want to target improving parent's beliefs about the importance of active play and alleviating the safety concerns they have about letting their child play outdoors. For many parents, structured and organized activities are seen as a means of providing their children with opportunities for

enrichment, which has led to busier schedules and less free time for play, an unstructured and unorganized activity that is often perceived as having little value (*Ginsburg, 2007*). Despite dwindling crime and injury rates, many parents perceive that it is unsafe for children to play outdoors unsupervised (*Carver, Timperio & Crawford, 2008b*; *Janssen, 2014*). For instance, an Australian study reported that ∼80% of parents of 10–12 year olds are concerned about strangers and automobile traffic in their neighborhood (*Timperio et al., 2004*). These perceived dangers are negatively associated with children's outdoor physical activity (*Carver, Timperio & Crawford, 2008b*; *Heitzler et al., 2006*; *Page et al., 2010*; *Veitch, Salmon & Ball, 2010*; *Wen et al., 2009*). The association between perceived safety and child physical activity may be mediated by independent mobility, which refers to the extent to which children are allowed to roam outdoors unsupervised. A limited independent mobility is associated with a lack of active play (*Prezza et al., 2001*; *Schoeppe et al., 2013*).

Strengths of our study include the specificity of measures (e.g., measures of parental encouragement and child physical activity that were specific to outdoor active play) and the consideration of individual, family, and neighborhood level covariates. This study is limited by the cross-sectional design which does not enable us to establish the temporal nature of the observed associations. Also, all of the data for this study was collected in late fall. Children's total physical activity differs across seasons (*Carson & Spence, 2010*), and it is possible that the relationships between parental encouragement and outdoor active play reported on here may have been different had the data been collected at another time of year. In addition, this study relied on parental-reports, which would have led to misclassification of the study variables, particularly the outdoor active play frequency score. It is possible that parents who encouraged play more frequently were particularly biased in their responses to the outdoor active play frequency questions, which would have led to differential misclassification and overestimated associations between parental encouragement and outdoor active play. Furthermore, we did not consider features of the physical environment (e.g., parks, playgrounds, green space) and it is possible that the observed relationships could have been confounded by these features. Finally, the observed associations could have been influenced by a common method variance (i.e., variance attributable to the measurement method and not the constructs the measures represent).

## CONCLUSION

These findings suggest that parental encouragement for outdoor play in the form of verbal cues is associated with outdoor active play. Future research in this topic area would benefit from using a longitudinal design and obtaining objective measures of physical activity. Intervention studies are also needed to determine effective approaches for increasing parental encouragement for outdoor play. It is hoped that the findings from this study will provide information that could be directed toward parents on strategies they could use to increase their child's physical activity.

### Funding

This research was supported by a Canada Research Chair award given to Ian Janssen. The funders had no role in study design, data collection and analysis, decision to publish, or preparation of the manuscript.

### Grant Disclosures

The following grant information was disclosed by the authors:
Canada Research Chair award.

### Competing Interests

The authors declare there are no competing interests.

### Author Contributions

- Thomas Ferrao conceived and designed the experiments, performed the experiments, analyzed the data, wrote the paper, prepared figures and/or tables.
- Ian Janssen conceived and designed the experiments, performed the experiments, contributed reagents/materials/analysis tools, reviewed drafts of the paper.

### Human Ethics

The following information was supplied relating to ethical approvals (i.e., approving body and any reference numbers):

This research was approved by the General Research Ethics Board at Queen's University (file # 6014210).

### Data Availability

The database file has been provided as a Data S1.

### Supplemental Information

Supplemental information for this article can be found online at http://dx.doi.org/10.7717/peerj.1463#supplemental-information.

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
