# Peer review of "Parental encouragement is positively associated with outdoor active play outside of school hours among 7–12 year olds"

_PeerJ, doi:10.7717/peerj.1463_

## Round 0.1 · original submission · Major Revisions

· Academic Editor

Major Revisions

Dear authors,

After carefully reading your paper and the comments made by the reviewers, I think your manuscript has can be published in PeerJ, once you have resolved the issues raised by both reviewers.

With respect and best regards,
Dr Palazón-Bru

Reviewer 1 ·

Basic reporting

In this article the authors examine outdoor play as a function of parents’ use of verbal cues in a large sample. In many respects, the study is a replication of earlier work on parental practices that may be related to youth physical activity. The new twist here is the focus on outdoor play specifically. The study is thorough in terms of the covariates that were included in the analyses.

However, the author’s research question, choice of covariates, and analytical approach do not appear to be driven by theory or a strong rationale. For example, why is encouragement the focus despite the inclusion of other relevant (and statistically significant) parental practices? Moreover, some of the variables included in the analyses are examined as direct effects, when theory may suggest they are exogenous to parental encouragement (e.g. parent risk perception). Finally, a strong theoretical approach may have steered the treatment of contextual factors. A social ecological perspective would suggest that a more appropriate approach is to model such factors as moderators of the main study variables.

In the introduction, prevalence data does not coincide with the country from which participants were selected. Statistics are reported for Canada, yet the respondents are from the US.

Reliability was not consistently reported for all measures. This is especially notable for the primary variable of interest, parental encouragement.

Experimental design

A major limitation is that all measures are based on cross-sectional data obtained from the parent's self-report. Thus, this study does not use methods that would be considered state-of-the-art.

Validity of the findings

The likely presence of common-method variance may have resulted in the overestimation of parameters.

Reviewer 2 ·

Basic reporting

No comments

Experimental design

No comments

Validity of the findings

No comments

Additional comments

The current study examined the association of parental encouragement with outdoor active plays outside school hours. The major findings were that parental encouragement was a strong, positive correlate of outdoor active play outside school hours. Overall, the manuscript is well-written and addressing an interesting topic with a simple study design. Personally, I appreciate the measures for possible covariates because they strengthen the quality of your study. However, while the current study addresses an important topic and is consistent with the scope of PeerJ, it is suffered from several constraints regarding methodology and the way results being interpreted. The authors should try to address the following issues before this manuscript can reveal its potential in adding to the knowledge base.

Title-
1. I will suggest the authors revise the title as “Parental encouragement is positively associated with outdoor active play outside school hours among children”. Given outdoor active plays “outside school hours” is the focus of your study, you could specify it on the title.

Abstract-
1. The abstract is easy-to-follow. I will only suggest the authors to add additional information regarding how the key measures were indexed (i.e., average scores, summation of scores) in Methods.

Introduction-
1. Generally, the Introduction is well-written. I agree with most of the ideas addressed. Yet, one issue I would like to raise is that the authors should provide more information regarding: 1) why study on the association between parental encouragement and active plays “outside school settings” so important? 2) what additional information can we get through this line of research? 3) how this line of research be helpful to parents and children?
2. Similarly, you may probably move the context within line 212-214 to the second paragraph in this section to strengthen your argument.
3. Another minor issue is that the authors raised the issue of the prevalence of physical inactivity among Canadian children at the beginning of Intro. However, given your articles will be read by readers around the world, additional information for the worldwide prevalence of physical inactivity among children may be helpful in providing a more comprehensive view.

Method-
1. Regarding the participants part, can the authors specify why parents completed the survey based on the “oldest” children if they have more than one.
2. Even though the children did not directly participate into the study, had they notified by their parents that their personal information would be revealed?
3. Did the authors collect demographic data of the parents such as gender and age? Since these variables might affect parents’ perception toward PA behavior of their children, additional information from the parents could be helpful in minimizing the effects of potential confounders.
4. I would like to see the authors address the indifference in the time children spending in PA in school settings. It may be possible that children who spend greater or fewer hours in PA in school settings affect their engagement in PA outside school hours.
5. Regarding covariates, I would like to know did participants in the current study had the same opportunity to access PA facilities such as parks, playgrounds, or forests? By now the measures included only urban forms, thus additional and detailed information regarding the living environment could be of help.
6. For the measures of outdoor active play and total physical activity outside of school, I would like to see the authors explain why parents were asked recall weekly frequency of PA during the past month. Does the recalls of PA behavior within the past month strong and solid enough to exclude any variations (e.g., the children got a flu, the children went for traveling)?

Discussion-
1. Personally I agree with most of the interpretations addressed; yet, my major concern leaves on the issue of overstatement. Given the entire study talks about only “association”, any implication for a “causal relation” should be avoided. Accordingly, the statement from line 217- 220 are somewhat exaggerated to me because the wordings imply a causal relation between parental encouragement and active plays.
2. As mentioned in the Intro section, I would like to see the authors give brief statement regarding how the current findings informative and helpful to the parents and children.

---

## Round 0.2 · accepted · Accept

· Academic Editor

Accept

Dear authors,

I have considered the new version of your work, and I think it has high standards to be published in PeerJ.

Congratulations!

With respect and best regards,
Dr Palazón-Bru (academic editor)

Reviewer 1 ·

Basic reporting

The paper is well written and the methods and results meet the criteria set forth by PeerJ

Experimental design

The study is limited in the extent to which we can infer a causal relationship between parental encouragement and children's active play. The reliance on parent self-report does not represent state-of-the-art methods.

Validity of the findings

The findings are reasonable given the analysis.

Additional comments

The authors have strengthened the paper in several ways in response to the first round of reviews. However, the paper could be strengthened further with the inclusion or discussion of relevant theory regarding the choice to examine parental encouragement, specifically. Multiple articles cited by the authors make a point to examine parental influence in the context of theory. It is still unclear why the authors, aside from a disciplinary argument, have omitted such discussion. Nevertheless, the paper offers strong and appropriately enough, encouraging results regarding the role parents may play in children's activity patterns.